# 7,8-Dihydroxiflavone Maintains Retinal Functionality and Protects Various Types of RGCs in Adult Rats with Optic Nerve Transection

**DOI:** 10.3390/ijms222111815

**Published:** 2021-10-30

**Authors:** Alejandro Gallego-Ortega, Beatriz Vidal-Villegas, María Norte-Muñoz, Manuel Salinas-Navarro, Marcelino Avilés-Trigueros, María Paz Villegas-Pérez, Manuel Vidal-Sanz

**Affiliations:** 1Departamento de Oftalmología, Instituto Murciano de Investigación Biosanitaria (IMIB) Virgen de la Arrixaca, Universidad de Murcia, Campus de CC de la Salud, El Palmar, 30120 Murcia, Spain; beatrizvidalvillegas@gmail.com (B.V.-V.); maria.norte@um.es (M.N.-M.); manuel.salinas@um.es (M.S.-N.); marcelin@um.es (M.A.-T.); mpville@um.es (M.P.V.-P.); 2Servicio de Oftalmología, Hospital Clínico San Carlos, Instituto de Investigación Sanitaria del Hospital Clínico San Carlos (IdIC), 28040 Madrid, Spain

**Keywords:** axotomy, intraorbital optic nerve transection, adult Sprague Dawley rat, retinal ganglion cells, 7,8-dihydroxiflavone, full field electroretinogram, scotopic threshold response (STR), Brn3a, osteopontin (OPN), Tbr2 (T-box transcription factor T-brain 2), α-RGCs, α-OFF RGCs, M4 ipRGC/α-ON-Sustained RGC

## Abstract

To analyze the neuroprotective effects of 7,8-Dihydroxyflavone (DHF) in vivo and ex vivo, adult albino Sprague-Dawley rats were given a left intraorbital optic nerve transection (IONT) and were divided in two groups: One was treated daily with intraperitoneal (ip) DHF (5 mg/kg) (*n* = 24) and the other (*n* = 18) received ip vehicle (1% DMSO in 0.9% NaCl) from one day before IONT until processing. At 5, 7, 10, 12, 14, and 21 days (d) after IONT, full field electroretinograms (ERG) were recorded from both experimental and one additional naïve-control group (*n* = 6). Treated rats were analyzed 7 (*n* = 14), 14 (*n* = 14) or 21 d (*n* = 14) after IONT, and the retinas immune stained against Brn3a, Osteopontin (OPN) and the T-box transcription factor T-brain 2 (Tbr2) to identify surviving retinal ganglion cells (RGCs) (Brn3a^+^), α-like (OPN^+^), α-OFF like (OPN^+^Brn3a^+^) or M4-like/α-ON sustained RGCs (OPN^+^Tbr^+^). Naïve and right treated retinas showed normal ERG recordings. Left vehicle-treated retinas showed decreased amplitudes of the scotopic threshold response (pSTR) (as early as 5 d), the rod b-wave, the mixed response and the cone response (as early as 10 d), which did not recover with time. In these retinas, by day 7 the total numbers of Brn3a^+^RGCs, OPN^+^RGCs and OPN^+^Tbr2^+^RGCs decreased to less than one half and OPN^+^Brn3a^+^RGCs decreased to approximately 0.5%, and Brn3a^+^RGCs showed a progressive loss with time, while OPN^+^RGCs and OPN^+^Tbr2^+^RGCs did not diminish after seven days. Compared to vehicle-treated, the left DHF-treated retinas showed significantly greater amplitudes of the pSTR, normal b-wave values and significantly greater numbers of OPN^+^RGCs and OPN^+^Tbr2^+^RGCs for up to 14 d and of Brn3a^+^RGCs for up to 21 days. DHF affords significant rescue of Brn3a^+^RGCs, OPN^+^RGCs and OPN^+^Tbr2^+^RGCs, but not OPN^+^Brn3a^+^RGCs, and preserves functional ERG responses after IONT.

## 1. Introduction

The rodent visual system has been used to explore the degenerative and regenerative potential of adult mammalian central nervous system neurons [1,2,3,4,5,6,7,8,9]. One popular experimental model consists of severing the entire visual pathway at its exit from the retina towards the brain, near the optic nerve (ON) head, and such a lesion by complete crush or transection leads to a permanent disconnection of the retina from its main target regions in the brain. Moreover, axotomy also results in a number of retrograde changes that affect several functional, morphological and molecular attributes [10], and most importantly it leads within a very short period of time to the loss of the vast majority of the axotomized population of RGCs [11,12,13]. Several strategies have been explored to slow down the loss of RGCS following several types of retinal insults, including the use of alpha two agonists [14,15,16,17,18,19,20,21] and the use of substances with neuroprotective properties, such as trophic factors and neurotrophins [22,23,24,25,26,27,28,29], of which BDNF is the most potent neuroprotectant for the retina [30], and the caspase inhibitors [31], among others [32]. Moreover, different pharmacological approaches have provided new evidence of RGCs protection [33,34,35].

More recently, 7,8-Dihydroxiflavone, a potent TrkB agonist that acts as a BDNF mimetic (for review see [36]) and crosses the blood brain barrier [37], has been shown to prevent axotomy-induced RGC loss in adult rats in vivo [38] through TrkB signaling and subsequent activation of two main intracellular downstream pathways: mitogen-activated protein kinase (MAPK)/ERK and phosphatidylinositol 3 kinase (PI3K)/AKT [39], that block both the intrinsic and extrinsic pathways of apoptosis [40]. However, at present we ignore whether the large proportion of RGCs rescued with DHF from axotomy-induced cell death maintain their functional properties.

There are many types of rat RGCs. A classification of the albino rat RGCs distinguished three morphological types of cells based on their dendritic and soma size: large (A cells), small (B cells) and medium that could be further subdivided (C cells) [41]. The A cells, also named α-RGCs (type I of Perry) [42] have in common large size (soma and dendritic size are among the largest of all RGCs), a fast conduction velocity (they convey the first light signals to the brain), mono-stratified dendritic arbors within different strata of the inner plexiform layer (IPL) [43,44,45], high neurofilaments content [46], and large receptive fields [45]. The α-RGCs represent a major input for image processing [47] and four types of α-RGCs have been described in rats [45] and mice [47], based on their response to light stimulus and the level of stratification of their dendritic trees in the IPL. Thus, according to their mono-stratification within the IPL, from inner to outer, the following types have been distinguished: ON-sustained, ON-transient, OFF-transient or OFF-sustained α-RGCs. In mice, α-RGCs express a secreted phosphoprotein, osteopontin (OPN) [47,48,49] and may be identified with OPN antibodies [47]. Moreover α-OFF RGCs may be identified with a combination of OPN and Brn3a antibodies [47].

Several types of melanopsin expressing intrinsic photosensitive RGCs (ipRGCs) have been described in mice (6; M1–M6) and in rats (5; M1–M5) [50,51,52]. Recent studies have shown in mice that all the ipRGC types express the T-box brain 2 (Tbr2), a T-box-containing transcription factor which may thus be used as a marker for ipRGCs [49,53,54,55]. One such type, the M4 ipRGC, has been shown to correspond with the α-ON sustained RGCs in rats [51] and mice [56,57,58]. In mice, M4 express OPN [47,49] and low levels of melanopsin and thus cannot be readily detected with standard immunohistochemical procedures against melanopsin [47,51,56,58,59]. These cells project mainly to the dorsal lateral geniculate nucleus and participate in image-forming visual functions [56,57], are the most sensitive RGC type to dim-light signals [60], play a major role in contrast detection [61] and, depending on their light adaptation state, show color-opponent responses [58]. It has been documented that these M4 cells correspond to the type I ipRGCs of early post-natal development [62]. In mice, M4 cells may be identified with the co-staining of OPN and Tbr2 antibodies [49,55].

The α-ON sustained RGCs (M4) and the three types of ipRGCs that contain high levels of melanopsin, which we will refer to, from now on, as melanopsin expressing RGCs (m^+^RGCs) (M1-M3), have been shown to exhibit a particular resilience to optic nerve or retinal injuries in adult pigmented mice [48,49], albino [38,63,64] and pigmented rats [65], although other reports have indicated that OFF-transient α-RGCs are very susceptible to optic nerve crush in mice [66]. However, whether albino rat M4-like and α-like RGCs are resilient to optic nerve injury and responsive to DHF-afforded protection is presently unknown.

Our present studies further extend our recent work, demonstrating, for the first time in vivo, the neuroprotective effects of 7,8-Dihydroxiflavone on adult injured rat retinal ganglion cells (RGCs) [38]. The present work has two main objectives: (i) to examine in vivo longitudinally retinal functionality with full field electroretinograms (ERG), to ascertain whether DHF-rescued RGCs are functional. Thus, we have examined vehicle- versus DHF-treated retinas using ERG to investigate alterations of the main ERG components, including the positive scotopic threshold response (pSTR), a-wave and b-wave, which are mediated largely by activity of the main retinal neuronal populations, the RGCs, photoreceptors, and bipolar cells, respectively [67,68,69], and; (ii) to further explore the responses of specific types of RGCs to injury and protection. RGCs may comprise up to 46 different types, some of which are known to differ in their response to injury and protection, and thus we examine the responses of the general population of RGCS, α-RGCs, α-OFF RGCs and M4 ipRGC/α-ON RCCs to axotomy and DHF afforded protection. Thus, we performed ex vivo analysis, at different survival intervals, to investigate the protective effects of DHF on the general population and several types of RGCs (α-like, α-OFF-like and M4-like), identified with Brn3a, OPN, OPN+Brn3a and OPN+Tbr2 antibodies, respectively.

We show that following IONT, there is: (i) significant, progressive and permanent diminutions of the scotopic threshold response, which are prevented in the DHF-treated group; (ii) significant diminutions of the b-wave of the rod response, the mixed and photopic response, which were also prevented with DHF-treatment; (iii) rapid and massive loss in vehicle-treated retinas of Brn3a^+^, OPN^+^, OPN^+^Brn3a^+^ (α-OFF like) and OPN^+^Tbr2^+^ RGCs, that is progressive for Brn3a^+^RGCs but not for the other types, and (iv) increased survival in the DHF treated group of Brn3a^+^RGCs, OPN^+^RGCs (α-like) and OPN^+^Tbr2^+^RGCs (M4-like), but not OPN^+^Brn3a^+^RGCs (α-OFF like).

## 2. Results

Following IONT and systemic vehicle- or DHF-treatment we have investigated the functional effects in the retina by recording full-field ERGs, and the fate of several types of RGCs using various molecular markers. Overall, our in vivo studies indicate that optic nerve injury results in reductions of the b-wave, as well as drastic and permanent reductions of the early component of the ERG, the positive scotopic threshold response, and we show that most of these functional alterations are prevented with systemic DHF treatment. Our ex vivo analysis shows that IONT induces massive and rapid loss of RGCs labelled with Brn3a^+^, OPN^+^, OPN^+^Brn3a^+^ or OPN^+^Tbr2^+^ that is progressive for Brn3a^+^RGCs but not for the other types, and this loss may be largely prevented with DHF treatment, except for OPN^+^Brn3a^+^RGCs.

### 2.1. Alterations of the Main ERG Waves

The IONT experimental groups, treated with vehicle or DHF, had simultaneous electroretinogram recordings in both eyes (the left IONT and its right fellow intact eye), and were registered longitudinally at 5, 7, 10, 12, 14 and 21 days after axotomy. In addition, both eyes from 6 naïve adult rats were also recorded simultaneously, analyzed and used as controls (*n* = 12). There were no differences between the mean waves recorded from naïve and the fellow right retinas of experimental groups (Figure 1 and Figure 2).

#### 2.1.1. Scotopic Threshold Responses

To assess the functional status of the RGC population, we analyzed the scotopic threshold response (STR), a wave that is recorded under special conditions stimulating the dark-adapted retina with very low light intensities (−4.4 log cd s/m^2^). This response provides a wave with two components, one positive (pSTR) and one negative (nSTR). In this work we have focused on the positive component of the wave that is directly related to the depolarization of the RGCs [67,70], and is measured from baseline to the highest point of the wave that was generated under normal conditions at approximately 130 ms after stimulus presentation (Figure 1). In our experimental vehicle- and DHF-treated groups the pSTR was reduced by 5 d to 40% when compared to control (*p* = 0.001, One-way Anova). In the vehicle-treated groups, there was a further reduction at 10 d (35%) with further progression at 21 d (17%), whereas in the DHF-treated groups, after the initial reduction at 5 d there was no further reduction until day 21 (45%). Mean pSTR wave amplitudes were significantly greater in the DHF-treated groups when compared to the vehicle-treated, at 7 (0.024 ± 0.007 mV, *n* = 24 vs. 0.016 ± 0.009 mV, *n* = 18; *p* ≤ 0.01 One-way Anova), 10 (0.022 ± 0.06 mV, *n* = 16 vs. 0.011 ± 0.002 mV, *n* = 12; *p* = 0.0005 One-way Anova), 12 (0.023 ± 0.005 mV, *n* = 16 vs. 0.011 ± 0.004 mV, *n* = 12; *p* ≤ 0.0001 One-way Anova), 14 (0.022 ± 0.004 mV, *n* = 16 vs. 0.009 ± 0.004 mV, *n* = 12; *p* ≤ 0.0001 One-way Anova) and 21 d after IONT (0.015 ± 0.002 mV, *n* = 8 vs. 0.004 ± 0.001 mV, *n* = 6; *p* = 0.0001 One-way Anova) (Figure 1).

#### 2.1.2. Rod Response

Scotopic recordings from rod bipolar cells were registered with significantly increased stimulus intensities up to −2.5 log cd s/m^2^ (Figure 2A). Naïve-control retinas (*n* = 12) showed a mean amplitude of 0.29 mV. The left eyes of DHF-treated animals showed no differences when compared to their right-contralateral naïve-control, at any time interval, implying that DHF-treatment contributed to the maintenance of full rod bipolar function and proper depolarization of rod bipolar cells. In contrast, the left eyes of the vehicle-treated rats showed a significant reduction of 18% at 10 days that further increased to 42% by 21 days (Figure 2A).

#### 2.1.3. Mixed Response

The study of the mixed response (obtained with stimulus intensities of the order of 0.5 log cd s/m^2^) allows analysis of the rapid hyperpolarization of the photoreceptors to light, reflected in the negative component (a-wave) and the depolarization of the bipolar cells, reflected in the positive component (b-wave). There were no differences between the DHF-treated and naïve-groups in the b-wave. However, in the vehicle-treated retinas, by 12 days after IONT, the b-wave showed a significant reduction of 35%, which was maintained until 21 days.

In the present study, there were no significant differences in the mean amplitude of the a-wave between the naïve, vehicle or DHF-treated groups (Figure 2B), indicating that the functionality of the photoreceptors after axotomy did not appear compromised in either group.

#### 2.1.4. Photopic b-wave

To assess the cone pathway exclusively, we adapted the animals to light for five minutes, recorded under photopic conditions (30 cd ambient light), and stimulated the retina with light pulses of 0.5 log cd·s/m^2^ intensity to obtain the photopic b-wave, which is thought to emerge by depolarization of retinal cone bipolar cells. As was the case under scotopic conditions, no reduction in the photopic b-wave was observed in the left retinas of DHF-treated animals, whereas there was a reduction of approximately 38% in the mean wave amplitude by 14 d after IONT in the left eyes of the vehicle-treated animals; this reduction was maintained until 21 d (Figure 2C).

### 2.2. RGC Survival at Different Time-Intervals

#### 2.2.1. Brn3a^+^ RGCs

The right retinas from vehicle- or DHF-treated groups were used as controls and showed normal appearance and topographic distribution of Brn3a^+^RGCs with mean total numbers of 81,085 ± 3056 (*n* = 42), which are in range with previous reports (Table 1, Figure 3A,B) [38,64,71,72,73]. The left IONT retinas showed a different general appearance depending on whether they belonged to the DHF- or vehicle-treated groups of rats. The main difference resided in the quantities of Brn3a^+^RGCs which, for similar survival intervals, were much greater in DHF- than in the corresponding vehicle-treated retinas. The left vehicle-treated retinas showed the typical diffuse loss of RGCs [63], and mean total numbers of Brn3a^+^RGCs decreased significantly at 7, 14 or 21 d after IONT, to approximately 45%, 30% or 10%, respectively, of the control values (Figure 3, Table 1). In contrast, and in agreement with our recent report [35], there were no differences at seven d after IONT (*p* = 0.114, Anova Test) between mean total numbers of Brn3a^+^RGCs in the left retinas of DHF-treated animals and their contralateral ones (Figure 3A). Moreover, the left DHF-treated retinas showed significantly greater numbers of surviving RGCs at all analyzed time intervals when compared to vehicle-treated ones (Figure 3A,C–E’; Table 1).

#### 2.2.2. OPN^+^RGCs

Analysis of retinal cross sections from 4 control retinas showed that of the OPN^+^RGCs, 36% were Brn3a^+^ (α-OFF like RGCs), 54% were Tbr2^+^ and 10% expressed neither (Figure 4). These proportions are somewhat similar to those obtained from 4 control retinas examined in whole-mounts showing that of the OPN^+^RGCs, 41% were Brn3a^+^ (α-OFF like RGCs), 55% were Tbr2^+^ and 4% expressed neither (Figure 5).

The right control retinas showed OPN^+^RGCs distributed throughout the retinas with higher densities in the inferior retina and maximum densities in the inferior-temporal retina (Figure 6). The mean total numbers of OPN^+^RGCs were 1917 ± 229 (*n* = 42) (Figure 4, Table 1). 

The left IONT vehicle-treated retinas showed by day seven a significant reduction in their mean total number of OPN^+^RGCs to approximately 38.4%, that did not decrease further with increasing survival intervals of 14 d (35.7%) (*p* = 0.002; Anova Test) or 21 d (35%) (*p* = 0.001; Anova Test) (Figure 6, Table 1). However, DHF-treated retinas, when compared to vehicle-treated retinas, showed significantly greater total numbers of OPN^+^RGCs at seven d (*p* = 0.0007; Anova Test) and 14 d (*p* = 0.0007; Anova Test), but not at 21 d (*p* > 0.99; Anova Test) after IONT (Figure 6, Table 1). When retinal wholemounts of experimental retinas were examined for the presence of OPN^+^Brn3a^+^RGCs (α-OFF like RGCs), these were rarely found with mean total numbers of 3 to 4 cells per retina, irrespective or their treatment or survival interval. Thus, OPN^+^Brn3a^+^RGCs represent ≈ 0.4% of the values observed in the contralateral retinas (Figure 7, Table 1).

#### 2.2.3. OPN^+^Tbr2^+^RGCs

The right retinas used as controls showed OPN^+^Tbr2^+^RGCs distributed throughout the retinas, but with higher densities in the inferior retina and maximum densities in the inferior-temporal retina. The mean total numbers of OPN^+^Tbr2^+^RGCs in control retinas were 1015 ± 120 (*n* = 42) (Figure 8, Table 1). There were hardly any OPN^+^Tbr2^+^RGC also labelled with Brn3a.

The left IONT vehicle-treated retinas showed seven d after IONT a significant reduction in their mean total number of OPN^+^Tbr2^+^RGCs (*p* < 0.0001, Anova Test) to approximately 26.5%, that did not decrease significantly further (*p* > 0.83, Anova Test) with increasing survival intervals of 14 d (22.9%) or 21d (19.9%) (Figure 9, Table 1). The left IONT DHF-treated retinas showed by seven d after IONT a significant reduction in their mean total number of OPN^+^Tbr2^+^RGCs to approximately 65.9%, that further decreased with increasing survival intervals of 14 d (33.6%) or 21 d (20.19%) (*p* < 0.0001, Anova Test) (Figure 9, Table 1). However, there were significantly greater total numbers of OPN^+^Tbr2^+^RGCs in the DHF- than in the corresponding vehicle-treated retinas at 7 and 14 days (*p* < 0.005, Anova Test), but not at 21 d after IONT (*p* > 0.99 Anova Test) (Table 1).

## 3. Discussion

Using full field ERG recordings to study the ERG components, we have examined longitudinally in vivo the functionality of DHF- versus vehicle-treated retinas against IONT-induced retinal degeneration. Ex vivo, we have analyzed the neuroprotective effects of DHF on axotomized adult rat RGCs, at different survival intervals, by identifying, counting and mapping Brn3a^+^, OPN^+^ (presumably α-RGCs), OPN^+^Brn3a^+^ (presumably α-OFF RGCs) and OPN^+^Tbr2^+^ (presumably M4 ipRGC/ON-sustained α-RGCs) RGCs. Our present studies show total numbers of OPN^+^RGCs and OPN^+^Tbr2^+^RGCs in adult albino rats that were previously unknown. Our results show that, following IONT: (i) in vehicle-treated retinas there were permanent and significant diminutions of the scotopic threshold response, an important ERG parameter related to RGC function, and significant diminutions in the b-wave, a major component of the ERG, which imply alterations of the inner nuclear layer of the retina; (ii) we show, for the first time, that DHF-treated retinas exhibit amplitudes that were improved for pSTR and normal for the b-wave, indicating that such functional abnormalities may be prevented with DHF-treatment; (iii) in vehicle-treated retinas there was a rapid, massive and progressive loss of Brn3a^+^RGCs. For OPN^+^RGCs or OPN^+^Tbr2^+^RGCs this loss was also rapid and massive but not progressive, suggesting that both these RGC types are somewhat resilient to axotomy-induced death, while OPN^+^Brn3a^+^RGCs diminished to approximately 0.5%, and (iv) DHF afforded a significant protection of Brn3a^+^RGCs, OPN^+^RGCs and OPN^+^Tbr2^+^RGCs but not OPN^+^Brn3a^+^RGCs. Brn3a^+^RGCs were rescued at all time intervals, but OPN^+^RGCs and OPN^+^Tbr2^+^RGCs were rescued at seven and 14 d, but not at 21 d.

### 3.1. DHF Neuroprotection Involves Function of Ganglion Cells and Second Order Neurons

Although ERG changes related to optic nerve injury have been previously documented [67,68,70,74], it was unknown if DHF-afforded rescue of axotomized RGCS also involved protection of retinal function. The scotopic threshold responses are thought to reflect RGC activity [75,76,77]. Our longitudinal analysis showed in vehicle-treated retinas a significant progressive and permanent reduction of the pSTR that likely reflects the massive loss of RGCs, and is in agreement with previous studies following optic nerve axotomy [67,68,74] and other types of retinal injury such as acute [70,78] or chronic [79,80,81] elevation of the intraocular pressure. In contrast, the DHF-treated retinas showed a much smaller reduction of the pSTR, which was first significantly smaller by day five, but did not progress further until d 21, suggesting that the larger population of rescued RGCs contributes to the greater amplitude of the pSTR, thus indicating that DHF not only protected RGCs against IONT-induced cell loss, but preserved their function.

Following IONT and vehicle-treatment there were also significant decreases in the amplitudes of the b-waves recorded under different conditions (scotopic, mixed and photopic) that first appeared significant by 10–14 days after IONT without further recovery. Somewhat similar decrements in b-wave amplitudes were reported following IONT, from this [67] and other [74,82] laboratories. The b-wave amplitude reductions could be explained by transient down-regulation of specific genes involved in phototransduction following IONT [83,84], but the lack of a-wave alterations recorded following IONT in both vehicle- or DHF-treated groups suggests alternative explanations. Indeed, ongoing studies suggest that, in addition to the loss of RGCs that follows IONT, other non-RGC neurons, including cone-bipolar cells and photoreceptors, may result affected long-term after injury [85]. It is of interest that in the DHF-treated retinas, the b-waves, recorded under different illumination conditions, did not show alterations and thus it is likely that DHF-treatment also prevented such a disfunction.

### 3.2. Albino Rat Retinal Distribution and Total Numbers of Brn3a, α-like, and M4-like RGCs

In control retinas, Brn3a^+^RGCs were distributed throughout the retina following the typical pattern; there were higher densities in the superior retina with maximum densities at the temporal side of the visual streak that runs along the nasotemporal axis, approximately one mm above the optic disc [71,80,86]. Total numbers of labelled Brn3a^+^RGCs in our control retinas are within range of previously reported data from this laboratory using similar techniques [38,64,71,72,73].

The topography of OPN^+^RGCs agree with previous studies showing higher densities of α-RGCs within the inferior temporal retina [43]. Indeed, in albino and pigmented rat, α-RGCs showed greater densities in the temporal retina with bigger densities of the OFF subtype of α-RGCs in the temporal side [43,44,45] (Figure 6, and Figure 7). Moreover, our results indicate that approximately 36–41% of the OPN^+^RGCs were also Brn3a^+^ and this is in line with the finding in mice that Brn3a was expressed in approximately one half of α-RGCs, the α-OFF RGCs [47]. Mean total numbers of OPN^+^RGCs in our control retinas were ≈1900 (1917 ± 229; *n* = 42). Thus, as total numbers of albino rat RGCs labeled with fluorogold applied to the optic nerve head are ≈84,700 [73,87], OPN^+^RGCs in albino rats constitute ≈ 2.2% of all RGCs, a number that is in range with previous estimations for α-RGCs; between 1–4% of the rat RGC population [43,44,45] and between 1.3% [49] and 3.7% [57] of the mice RGC population. Our numbers are somewhat greater than the total numbers of α-RGCs reported by Dreher and colleagues [43] after massive HRP injections into retinorecipient nuclei (876) or neurofilament staining (791), but such differences may be related to the techniques employed to identify these retinal neurons.

In our studies, the mean total numbers of OPN^+^Tbr2^+^RGCs in control retinas were ≈ 1000 (1015 ± 120; *n* = 42), a figure in agreement with our observations in cross sections and wholemounts of four different naïve albino rat retinas showing that approximately one half of the OPN^+^RGCs are also Tbr2^+^ (Table 1, Figure 4 and Figure 5), and thus are M4-like ipRGCs. In control retinas OPN^+^Tbr2^+^RGCs showed a gradient distribution on the naso-temporal axis, with highest densities in the inferior temporal side of the retina. This finding is in agreement with mice M4 exhibiting a gradient across the nasal-temporal axis of the retina but disagrees in that mice M4 are more densely distributed in the superior temporal retina [43,50,58,59,88].

Although rat α-RGCs have been shown to differ from those of other mammals, including mice [43], our total number of OPN^+^Tbr2^+^RGCs in albino rats is slightly bigger than M4 ipRGCs in mice, which has been estimated to be about 850 M4 ipRGCs, approximately one half of the SMI-32 alpha RGCs [57]. However, a recent study in mice using melanopsin expression and RBPMS as a pan-RGC marker found that melanopsin-expressing cells account for approximately 6% of all RGCs, while Tbr2^+^RGCs account for approximately 12% of all RGCs [55]. Considering total numbers of mice RGCs labelled from the ON head in the order of ≈42,000 (42,658 ± 1540; *n* = 10 [69]), this would imply that total numbers of M1-M3 ipRGCs would equal 2520, while M4-M6 ipRGCs would account for a similar figure. Further studies are required to find out the exact numbers and proportions of each of the ipRGC types.

### 3.3. Resilience of α-RGCs and M4-like ipRGCs to IONT

The rodent retina consists of over 46 different RGC types [49], and specific types of RGCs may have idiosyncratic responses to injury and protection [38,48,49,89,90,91]. For example, a particular resilience has been documented for the melanopsin expressing ipRGCs against optic nerve injury [27,38,63,92,93,94], excitotoxicity induced by intravitreal injection of NMDA [64,95,96], acute [29] or chronic [28,65] ocular hypertension, or light induced phototoxicity [97]. Our results suggest that in order of sensibility, OPN^+^Brn3a^+^ (α-OFF like) are the most sensible, followed by OPN^+^Tbr2^+^RGCs (M4 like) which are slightly more sensible than OPN^+^RGCs (α-like) and Brn3a^+^RGCs to IONT. Indeed, by seven d after IONT, the loss of OPN^+^Brn3a^+^ is ≈99.5% compared to ≈74% OPN^+^Tbr2^+^RGCs, ≈62% OPN^+^RGCs and ≈55% Brn3a^+^RGCs (Table 1). However, OPN^+^RGCs and OPN^+^Tbr2^+^RGC loss did not progress between seven and 21 days after IONT, the latest time point examined in the present studies, so that by 21 d there is a relative increased survival of α- and M4-like RGCs when compared to Brn3a^+^RGCs; 25% and 20% versus 11%, respectively.

Overall, these results suggest that OPN^+^RGCs and OPN^+^Tbr2^+^RGCs are more resilient than Brn3a^+^RGCs, with OPN^+^Brn3a^+^RGCs being the most sensible ones. Somewhat similar results were reported in a recent study in adult pigmented rats, showing high susceptibility of alpha RGCs (identified with SMI-32) to intraorbital optic nerve crush [65]. In adult mice, it was found that α-RGCs were resilient to optic nerve crush [48], and among the four α-RGCs the two transient types were more labile than the sustained types [49,66], with α-OFF transient RGCs being most sensible [66]. It is possible that the particular resilience of both OPN^+^RGCs and OPN^+^Tbr2^+^RGCs is due to the expression of OPN, which is thought to be a RGC survival mediator [48].

### 3.4. DHF Prevents IONT-Induced RGC Loss

Our results indicate that as for the OPN^+^RGCs (α-like) the OPN^+^Tbr2^+^RGCs (M4-like) are also amenable to DHF-rescue, although this effect is transient and tappers off by 21 days. Our recent study documented that following IONT, systemic administration of DHF resulted in significant rescue of Brn3a^+^RGCs and in significant and permanent rescue of melanopsin expressing RGCs (m^+^RGCs) [35]. Here, we have examined the responses of additional types, the OPN^+^RGCs, presumably α-RGCs, the OPN^+^Brn3a^+^RGCs, presumably α-OFF RGCs and the OPN^+^Tbr2^+^RGCs, presumably ipRGC M4 (α-ON-sustained RGCs), to IONT and its response to DHF-afforded neuroprotection. Our results confirm our previous observations on the rescue of Brn3a^+^RGCs, and add new data indicating that α-like RGCs and M4 ipRGC/α-ON-S RGC respond to DHF. Indeed, DHF-treatment resulted in a substantial rescue of both types at seven and 14 days after IONT that tapered off by 21 d later, indicating their response to neuroprotection. However, OPN^+^Brn3a^+^RGCs (α-OFF like RGCs) were non-responsive to DHF treatment.

Our present results also suggest a different response of the M4 type of ipRGCs when compared to our recent study characterizing the response of the m^+^RGCs, which mainly account for the M1-M3 types of ipRGCs. The m^+^RGCs, following a transient downregulation of melanopsin, showed a survival of 41% at 14 d without further decay up to 60 d, and significant protection with DHF that resulted in a survival of 52% at 14 d that was permanent up to 60 d [35]. In contrast, the M4-like in the present studies showed a greater susceptibility to IONT that resulted in the survival of 23% at 14 d, although they also responded to DHF treatment resulting in transient protection with the survival of 34% by 14 days.

### 3.5. Possible Mechanisms of Action of DHF

DHF has been shown to exert a number of protective functions, including anticancer [98] and anti-oxidative stress [99], but the finding that it is a high-affinity agonist of the tropomyosin-related kinase B (TrkB) receptor of brain-derived neurotrophic factor (BDNF) represented a milestone. Furthermore, added to its capacity to act as a BDNF mimetic, DHF was found to have several pharmacokinetic properties that make it advantageous for its use as neuroprotectant, and thus its potential has been explored experimentally in a number of BDNF-related neurodegenerative diseases (for review see Table 1 of [36]). However, until very recently, it was unknown if DHF could afford protection of adult rat RGCs in vivo [38]. Indeed, previous in vitro and in vivo studies showing selective activation of TrkB prompted us to investigate in vivo the neuroprotective effects of DHF against optic nerve transection, a classic model of retinal injury that induces RGC degeneration but may be prevented with BDNF [30]. Recent studies have shown that DHF may prevent axotomy-induced RGC loss, an effect mediated through TrkB activation [39] involving the two main TrkB signaling pathways implicated in cell survival, PI3K/AKT and MAP/ERK [39].

However, whether DHF neuroprotection in the retina may also involve other TrkB signaling pathways such as PLC-γ and GTP-ases remains to be further investigated [39]. Moreover, DHF has been shown to have antioxidant activity and protect against glutamate induced excitotoxicitiy [100] or intravitreal injection of NMDA (Gallego-Ortega et al., unpublished observations), but whether this protection involves the same TrkB intracelular signalling pathways needs to be further investigated. Finally, optic nerve injury involves an inflammatory response in the injured and contralateral retina that is microglia mediated [32,101] and DHF has been shown to exert anti-inflammatory effects through microglia activation [102,103], thus it will be of interest to further correlate the role of microglial responses to fully understand the mechanisms of action of DHF.

### 3.6. Translational Relevance

Our present results further support DHF as a promising compound that could eventually be translated from bench- to bedside in retinal conditions involving RGC degeneration. Some of the most important injuries or diseases of the retina leading to RGC degeneration, such as glaucomatous optic neuropathy, optic nerve lesions, transient ischemia of the retina, acute ocular hypertension and others, involve common mechanisms of RGG death, including axotomy, excitotoxic damage and immune-inflammatory responses. As such, DHF has been documented to prevent oxidative stress [99], a common mechanism of cell death in retinal injuries or diseases. Moreover, DHF is a selective TrkB agonist and a potent BDNF mimetic [104], with proven in vivo protection of RGCs against axotomy [38] (and the present studies). Moreover, DHF has anti-inflammatory effects though microglia activation [102,103]. A number of pharmacokinetic properties make DHF advantageous for its use as a neuroprotectant, including its capacity to cross the blood brain barrier when administered systemically [37], its long and potent activation of the TrkB receptor [105] and its apparent lack of toxicity when administered chronically in vivo [38,99,105,106]

An overview of experimental studies using DHF as a neuroprotectant against neurological disorders (for review see Table 1 of [36]) reveals that: (i) a large majority of the studies have administered DHF chronically, although some have explored the effects of a single administration of DHF; (ii) most of the studies use a dose of 5 mg/kg, and (iii) the most frequent route for systemic administration was intraperitoneal, but per os and subcutaneous administrations were also frequent. Nevertheless, for the retina it would be interesting to study the possibility of using DHF administered topically, and this requires further investigations.

### 3.7. Present Limitations

Our functional assessment of the retinal responses following optic nerve injury and protection with DHF administered systemically relies on the recording of the main ERG waves, and these reflect electrical activity of large populations of neurons after a light stimulus. Thus, our assessment provides indirect evidence for the wellbeing of large number of retinal neurons contributing to each of the main recorded components. Further studies using additional stimulating parameters could help discern the main ON- and OFF-pathway alterations following IONT [107].

In the absence of markers to readily identify rat α-RGCs or M4 ipRCCs/α-ON-sustained RGCs [51], we assume for this study that in the adult albino rat, as shown for adult mice, α-RGCs may be identified with Osteopontin [47,49], α-OFF RGCs with the co-labelling of OPN and Brn3a, and the M4 ipRGCs with the co-labeling of OPN and Tbr2 [47,49,55]. However, immunohistochemical markers need to be used with caution as most are not entirely specific for a particular RGC type, and one cannot be absolutely certain of the true expression of a protein following retinal injury [63,83,84,91]. Nevertheless, Brn3a, OPN and Tbr2 have been shown to behave as vital markers for RGCs following retinal injuries. For instance, Brn3a has been shown to be a vital marker for RGCs after optic nerve lesion [71,94], Tbr2 is essential for the survival of adult Tbr2 ipRGCS [53,55] and OPN has been shown to be associated with highly resistant RGCs after injury [48].

## 4. Materials and Methods

### 4.1. Animals

For the present studies we have used adult female Sprague-Dawley rats (180 g) (animal house; University of Murcia) kept in temperature-controlled rooms with light/dark (12 h/12 h) cycles and food and water provided ad libitum. Animal manipulations were approved by the University of Murcia ethical animal studies committee (Codes: A13171103, A13170110 and A13170111), and followed the ARVO Statement for the Use of Animals in Ophthalmic and Vision Research, the European Union and the Spanish directives for the use of animals in experimental research. Animal manipulations involving pain or discomfort were produced under general anesthesia with a mixture of intraperitoneal ketamine (60 mg/Kg bw, Ketolar; Pfizer, Alcobendas, Madrid, Spain) and xylazine (10 mg/kg Rompun; Bayer, Kiel, Germany).

### 4.2. Axotomy of the Left Optic Nerve

The left optic nerve was transected intraorbitally, at approximately 0.5 mm from the ON head following previously described methods that are standard in this Laboratory [108]. In brief, to access the ON at the back of the eye, the skin over the superior orbital reem was divided and we dissected the contents on the supero-external aspect of the orbit, the external rectus and superior muscles were divided, the dura mater around the ON was divided longitudinally and the ON was transected entirely. Care was taken to avoid damage to the retinal blood supply, which runs on the inferior aspect of the dura mater [109,110]. At the end of the procedure, careful inspection of the eye fundus directly through the operating microscope ensured preservation of retinal blood supply.

### 4.3. Experimental Design and Drug Administration

Rats were divided in four groups. A first group (DHF; *n* = 24) was treated daily with ip DHF (5 mg/kg) starting 12 h before IONT. A second group (Vehicle; *n* = 18) was treated daily ip with vehicle (1% DMSO in 0.9% NaCl) starting 12 h before IONT. A third group (Naïve; *n* = 6) of intact rats was used to obtain ERG control parameters and were returned to the animal house. A fourth group consisted of four intact rats sacrificed to examine their retinas in wholemounts (*n* = 4) or cross sections (*n* = 4) and investigate the proportions of OPN^+^RGCs co-expressing Tbr2 or Brn3a.

For the present studies we administered DHF at a dose of 5 m/kg, because in our previous study [38] we performed a dose-response study for DHF and found that treatment with DHF at 4 or 5 mg/kg ip had a significant neuroprotective effect against axotomy-induced RGC loss, while treatment with DHF at a dose of 1, 2, 10 or 25 mg/kg did not show significant RGC rescue effects. Moreover, treatment with 5 mg/kg resulted in the survival of ≈95% RGCs at seven days, and thus this dose was considered optimal.

The DHF- and Vehicle-treated groups were examined longitudinally at 5, 7, 10, 12, 14 and 21 days after IONT to record flash electroretinogram (ERG) scotopic threshold response (STR), rod response, mixed response (a- and b-wave amplitudes) and photopic b wave (see below). For the histological analysis of the survival of different RGC types, eight rats from the DHF group and six rats from the vehicle group were sacrificed at 7 d, 14 d or 21 d after axotomy (Figure 9).

### 4.4. Electroretinography (ERG)

Longitudinal ERG recordings were registered simultaneously from both eyes of the same rats as follows: (i) at five and seven days (DHF *n* = 24; Veh *n* = 18) (ii) at 10, 12 and 14 days (DHF *n* = 16; Veh *n* = 12), and; (iii) at 21 days (DHF *n* = 8; Veh *n* = 6). For the electroretinographic study we used previously described methods in our laboratory [67,68,70,111]. In brief, after dark adaptation for 12 h, animals were anaesthetized, both eyes were dilated with topical mydriatic (Tropicamida 1%^®^, Alcon-Cusí, S.A., El Masnou, Barcelona, Spain) and electroretinogram (ERG) recordings were performed under red light, while animals were kept on a heating pad at 37 °C. Using a Ganzfeld dome capable of producing different light stimuli for retinal stimulation, different electroretinographic waves were recorded. The waves were recorded binocularly by means of a corneal lens (Burian-Allen electrode; Hansen Ophthalmic Development Laboratory, Coralville, IA, USA), a reference electrode placed in the mouth, and grounding at the base of the tail. Between the corneal lens and the cornea, we instilled methyl cellulose (Methocel 2%^®^; Novartis Laboratories CIBA Vision, Annonay, France) to improve conductivity. ERG signals were amplified and band-filtered between 0.1 and 1000 Hz (Four-Channel Differential AC Amplifier Model 1700, A-M Systems Inc, Carlsborg, WA, USA). Electrical signals were digitized at 20 kHz with a power laboratory data acquisition card (AD Instruments, Chalgrove, UK). RGC-mediated responses were recorded with light flashes ranging from −4.4 log cd s/m^2^ with dark-adapted mice. Rod-mediated responses were recorded at intensities of −2.5 log cd s/m^2^. Mixed rod- and cone-mediated signals in response to light flashes of 0.5 log cd s/m^2^. For the recording of cone-mediated responses, 0.5 log cd s/m^2^ light flashes were applied on a 30 cd/m^2^ rod-saturated background. The standard waves of the ERG were analyzed following guidelines of the International Society for Clinical Electrophysiology of Vision (ISCEV).

### 4.5. Animal Processing and Immunocytochemistry to Identify Different RGC Types

Because there are no clear markers for rat α- or M4-RGCs, we have used a combination of antibodies that in mice has proven useful to identify these RGC types. In mice, OPN and Tbr2 have been shown to behave as pan-markers for α-RGCs [47,48] and ipRGCs [53,55], respectively, and a combination of Tbr2 and OPN identifies α-ON-sustained/M4 ipRGCs [55], while a combination of OPN and Brn3a identifies α-OFF RGCs [47]. In addition, co-expression of OPN, Brn3a and Tbr2 were examined in additional control adult albino rat retinas in cross sections (*n* = 4) or wholemounts (*n* = 4).

Control (*n* = 4) or experimental rats following survival intervals of seven (*n* = 14), 14 (*n* = 14) or 21 (*n* = 14) days, were deeply anaesthetized with an i.p. overdose of barbiturate (Dolethal, Vetoquinol^®^, Especialidades Veterinarias S.A., Madrid, Spain) and perfused briefly with 0.9 NaCl and slowly with 4% paraformaldehyde in 0.1 M PBS (phosphate buffer saline). The eyes were dissected taking care to mark the superior pole with a 6/0 silk thread, four eye balls were further processed to obtain cryostat 16 µm thick retinal cross sections in the naso-temporal axis of the eye while the remaining retinas were prepared as flattened wholemounts, postfixed for an hour and rinsed in PBS [64]. Immunodetection followed previous protocols described in our Lab for retinal cross sections [86,112,113] or whole-mounted retinas [64,114].

Both retinas from experimental (left IONT and contralateral intact right) and control rats were incubated overnight in a PBS 2% Triton X-100 (Tx) solution containing primary antibodies against Brn3a, (mouse anti-Brn3a, 1:500 dilution, MAB1585 Millipore), Osteopontin (OPN) (goat anti-Osteopontin, 1:1000 dilution, AF808 Biotechne) and Tbr2 (rabbit anti-Tbr2, 1:1000 dilution, AB23345 Abcam).

After incubation of primary antibody, the retinas were rinsed in PBS 0.5% Triton X-100 (Tx) and incubated for 2 h with a mixture of secondary antibodies (1:500 in PBS-2% Tx, goat anti mouse Igg1 Alexa 555 (A21127), donkey anti rabbit Alexa 488 (A21206) and donkey anti goat Alexa 647 (A32849) Molecular Probes Thermo-Fisher, Madrid, Spain), rinsed in PBS, mounted vitreal side-up with antifading solution on subbed slides and covered with a coverslip sealed with nail polish. Cryostat cross sections that contained the ON head were also incubated as above, covered with antifade mounting media with DAPI and sealed with nail polish. Slides were kept in the refrigerator until fluorescence microscopy examination.

### 4.6. Assessment of Retinal Ganglion Cell Types

Image reconstructions of flattened immunolabelled retinas were obtained as described [70]. In brief, using an epifluorescence microscope (Leica DM6-B; Leica Microsytems, Wetzlar, Germany), the retinas were photographed (×10) in a raster-scan pattern without overlap or gap between images. Individual images were focused before acquisition, and obtained with the same focus under specific filters that allow identification of Brna3a^+^RGCs, OPN, and Trb2 cells, respectively.

Wholemount reconstructs were processed to obtain automatic quantification of Brn3a^+^RGCs for each retina, as described [64,115]. The OPN^+^, OPN^+^Tbr2^+^ and OPN^+^Brn3a^+^ cells were dotted manually on the photomontages and quantified with the aid of the graphic editing software Adobe Photoshop CS8.01 (Adobe Systems, Inc., San José, CA, USA) following standard methods in our laboratory [73]

The distribution of RGCs within each retina was examined with isodensity maps for Brn3a^+^RGCs or neighbor maps for the populations of OPN^+^Tbr2^+^, OPN^+^ or OPN^+^Brn3a^+^ cells, respectively [73,115]. Isodensity maps were built using the Brn3a^+^RGC densities obtained within each individual frame and presented with a color scale from 0 (purple) to ≥2500 (red) RGC s/mm^2^. OPN^+^Tbr2^+^, OPN^+^ or OPN^+^Brn3a^+^ cells neighbor maps represent their retinal distribution with each dot representing individual cells with a color that reveals the number of neighboring OPN^+^Tbr2^+^ or OPN^+^ cells within a radius of 0.276 mm from 0 to 4 (purple) to ≥32–35 (red) neighbors. Both neighbor and isodensity maps were made with Sigmaplot (SigmaPlot 9.0 for Windows; Systat Software, Inc., Richmond, CA, USA), as described [35].

In three cross-sections containing the optic disk (from the dorsal, central and ventral aspect of the optic disk) from each of the four control retinas, we counted the number of cells labelled with OPN alone, or that were double labeled with Tbr2 or Brn3a in fluorescence micrographs. These were obtained from each section (four from the nasal and four from the temporal, measuring 570 × 570 μm) and positioned at approximately 25%, 50%, 75% and 95% distance from the optic disk towards the periphery, as described [116,117]. Images were further processed with Adobe^®^ Photoshop^®^ CS 6 (Adobe Systems, Inc., San Jose, CA, USA) when needed.

### 4.7. Statistical Analysis

Data were analyzed and plotted with GraphPad Prism v.8 (GraphPad, San Diego, CA, USA). Averaged data is presented as mean ± standard deviations (SD). The statistical comparisons were done with GraphPad Prism v.8. Brown-Forsythe. One-way ANOVA was used when comparing more than two groups and Welch’s t-test when comparing two groups only. Differences were considered significant when *p* < 0.05.

## 5. Conclusions

Overall, our results indicate that IONT induces in vehicle-treated retinas dramatic permanent alterations of the pSTR, suggesting a functional impairment of the RGCs, the main contributors to this wave of the ERG. Such a functional impairment was prevented in the DHF-treated retinas, which resulted in functional as well as anatomical protection of this population. It is of interest that DHF-treatment also prevented the alterations recorded in the b-waves of the vehicle-treated retinas. Thus, it is possible that DHF not only rescued RGCs but prevented further injury to the inner nuclear layers that are mainly responsible for the generation of the b-wave. Surviving RGCs were evaluated structurally ex vivo by identifying, counting and mapping Brn3a, OPN, OPN-Brn3a and OPN-Tbr2 expressing neurons in the RGC layer. Our present studies show that in vehicle-treated rats, IONT resulted in the loss of ≈ 99.5% of the OPN^+^Brn3a^+^RGCs, in massive progressive loss of Brn3a^+^RGCs, and in massive but not progressive loss of OPN^+^RGCs and OPN^+^Tbr2^+^RGCs, suggesting a certain resilience of these RGC types to axotomy. Moreover, DHF-treatment in adult rats with IONT resulted in significant protection for the Brn3a, OPN and OPN-Tbr2 RGCs, suggesting that these populations are responsive to DHF-induced neuroprotection. However, OPN^+^Brn3a^+^RGCs were non-responsive to DHF treatment.

## Figures and Tables

**Figure 1 ijms-22-11815-f001:**
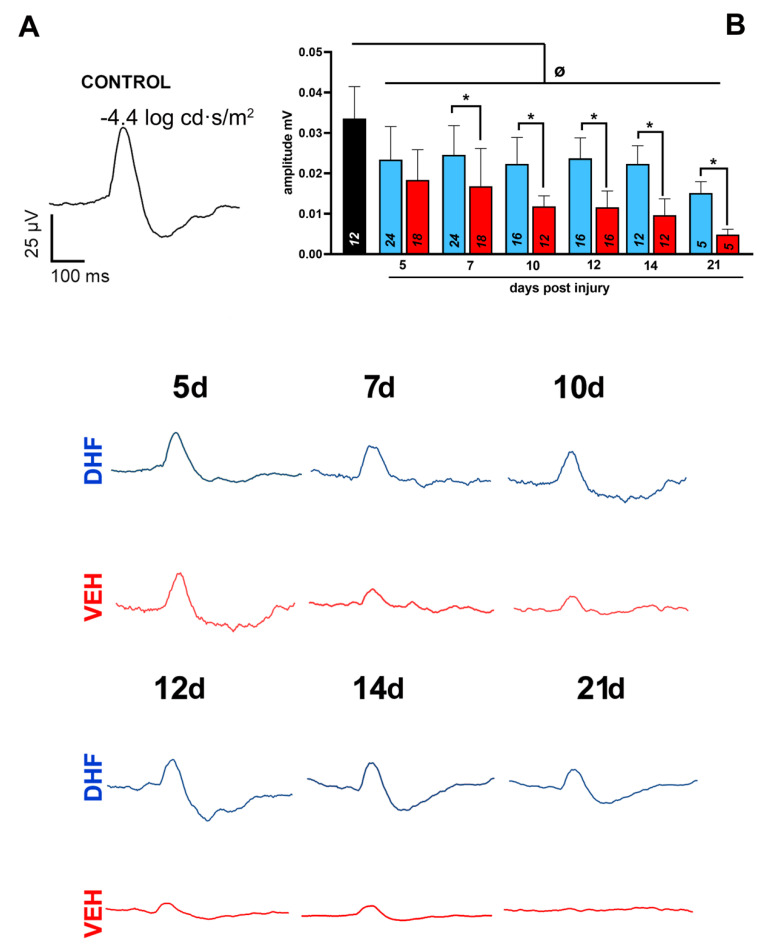
Treatment with 7,8-Dihydroxiflavone attenuates axotomy-induced RGC function loss. Adult albino rats were treated daily with i.p. vehicle (VEH) (1% DMSO in 0.9% NaCl) or DHF (5 mg/kg) starting the day before left optic nerve section. STR was recorded longitudinally in both eyes at days 5, 7, 10, 12, 14, and 21 after axotomy with a stimulus intensity of −4.4 log cd s/m^2^. (**A**) Representative traces of the STR recordings at different time intervals. Representative traces from a control (in black) or experimental retinas treated with DHF (in blue) or Vehicle (in red). (**B**) Bar histogram showing mean (±SD) amplitudes in millivolts of the control (in black, *n* = 12) and experimental groups treated with DHF (in blue) or VEH (in red) analyzed longitudinally at increasing survival intervals (*n* for each experimental group analyzed at different days post injury are indicated in the corresponding bar of the histogram). Differences between Control and DHF 7 d (*p* = 0.027); Differences between DHF 7 d and DHF 21 d (*p* = 0.0006 One-way ANOVA). One-way ANOVA * *p* < 0.05, differences between DHF and VEH treated group. One-way ANOVA ^Ø^ *p* < 0.05, differences between Control and the experimental groups.

**Figure 2 ijms-22-11815-f002:**
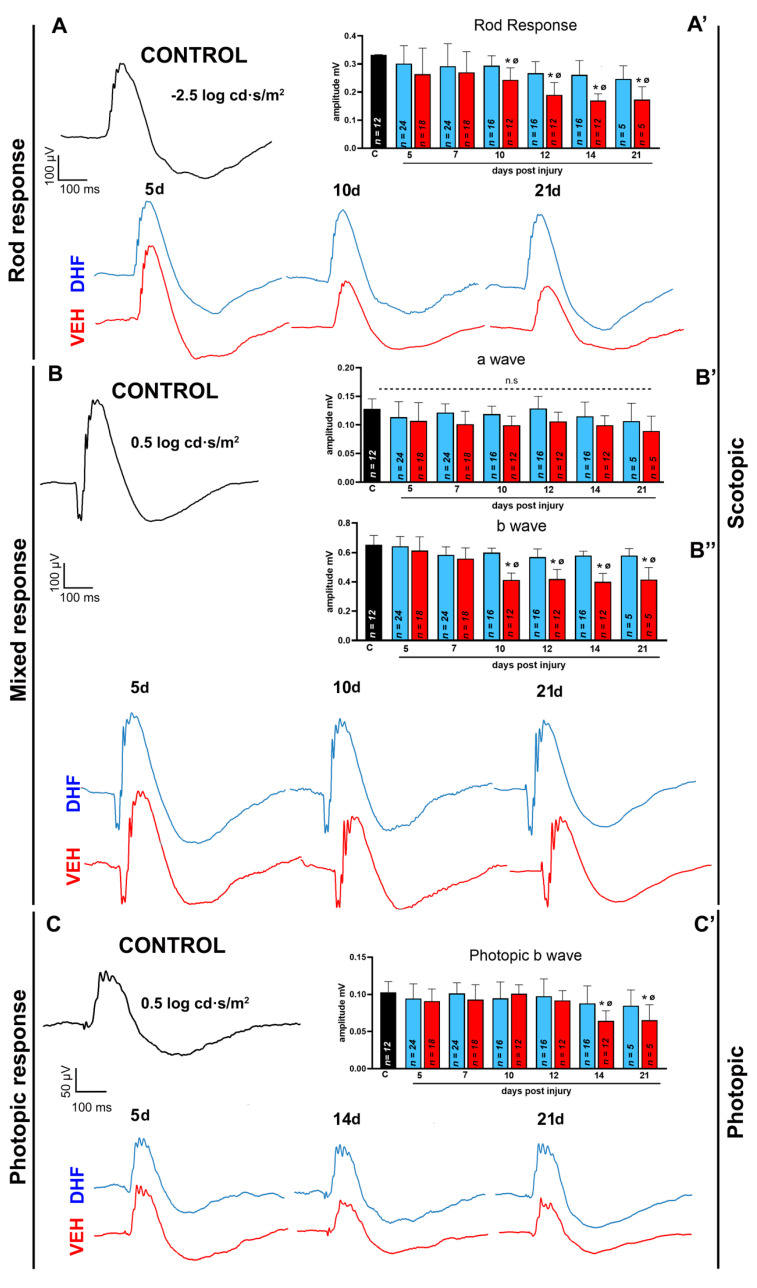
7,8-Dihydroxiflavone fully prevents function loss of second order retinal neurons. Adult albino rats were treated daily with i.p. vehicle (VEH) (1% DMSO in 0.9% NaCl) or DHF (5 mg/kg) starting the day before left optic nerve section. Rod response (−2.5 log cd s/m^2^), a wave (0.5 log cd s/m^2^), b wave (0.5 log cd s/m^2^) and photopic b wave (0.5 log cd s/m^2^) were recorded longitudinally in both eyes at 5, 7, 10, 12, 14 and 21 after the insult. (**A**–**C).** Representative traces of the recordings at different time intervals, from a control (in black) or experimental retinas treated with DHF (in blue) or Vehicle (in red) at different time points. (**A**) Representative traces of the rod response. (**B**) Representative traces of the mixed response (a-wave and b-wave). (**C**) Representative traces of the photopic response. (**A’**,**B’**,**B’’**,**C’)** Bar histogram showing mean (±SD) amplitudes in millivolts of the control (in black, *n* = 12) and experimental groups treated with DHF (in blue) or VEH (in red) analyzed longitudinally at increasing survival intervals (*n* for each experimental group analyzed at different days post injury are indicated in the corresponding bar of the histogram). One-way ANOVA * *p* < 0.05, differences between DHF and VEH treated group. One-way ANOVA ^Ø^ *p* < 0.05, differences between control and the experimental groups, n.s.: not significant.

**Figure 3 ijms-22-11815-f003:**
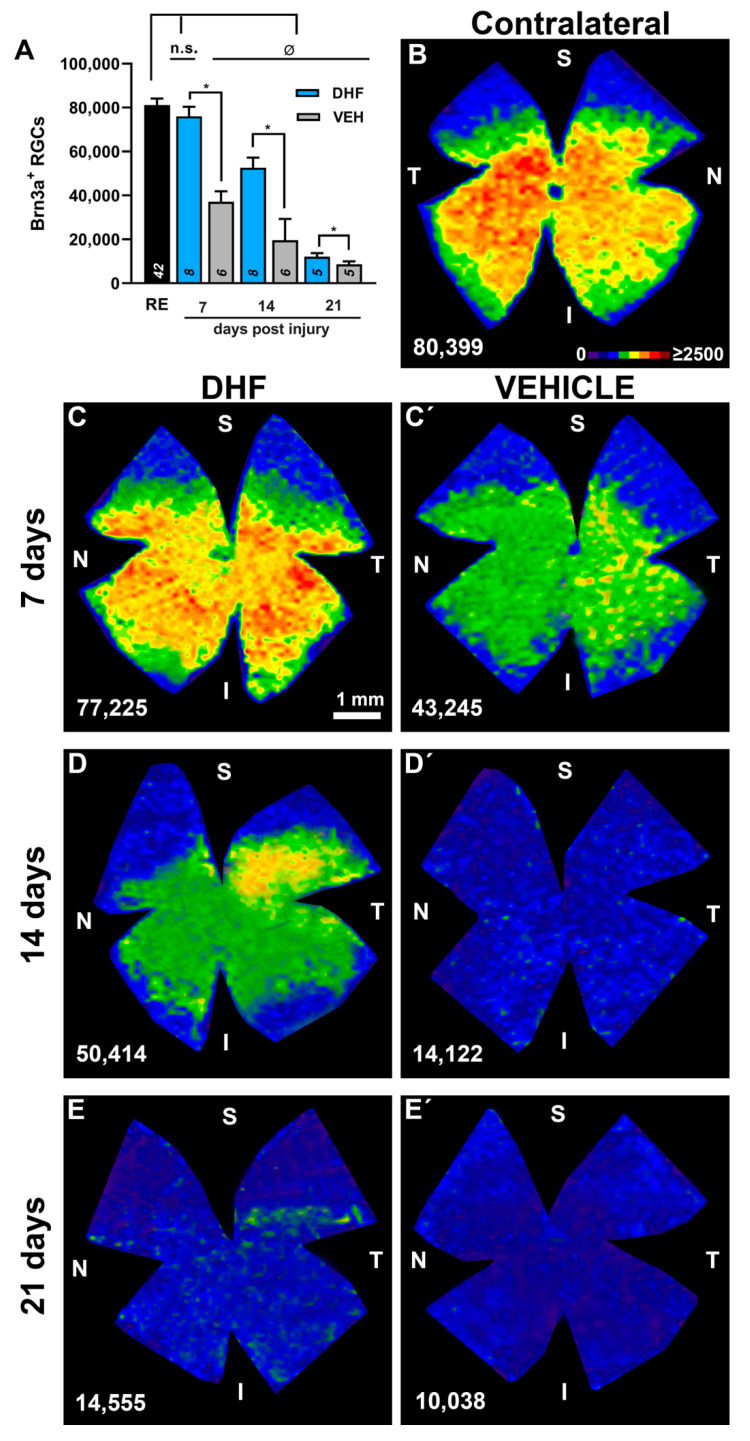
DHF prevents IONT-induced Brn3a+RGC loss. In adult albino rats, the left optic nerve was intraorbitally transected and animals were treated i.p. daily with vehicle (VEH) (1% DMSO in 0.9% NaCl) or DHF (5 mg/kg) starting one day before IONT until processing and analyzed at 7, 14 or 21 days post injury to determine Brn3a+RGC survival. (**A**) Bar histogram showing mean total numbers of Brn3a+RGCs in right fellow (RE) and left experimental retinas of rats treated with DHF or VEH (n for each experimental group analyzed at different days post injury are indicated in the corresponding bar of the histogram). One-way ANOVA * *p* < 0.05, differences between DHF and VEH treated group. One-way ANOVA ^Ø^ p < 0.05, differences between control and the experimental groups. (**B**–**E’**) Representative isodensity maps showing the topographical distribution of Brn3a+RGCs in an untouched contralateral retina (**B**), and in retinas treated with DHF (**C**–**E**) or VEHICLE (**C’**,**D’**,**E’**), analyzed at 7 (**C**,**C’**), 14 (**D**,**D’**) or 21 (**E**,**E’**) days. Isodensity maps illustrate the neuroprotective effects of DHF over the population of Brn3a+RGCs when compared to vehicle. Below each map the total number of Brn3a+RGCs is indicated for that retina. S: superior, N: nasal, I inferior, T, temporal. Isodensity map color scale from 0 RGCs/mm2 (purple) to ≥2500 (red) RGCs/mm2. Scale bar 1 mm, n.s.: not significant.

**Figure 4 ijms-22-11815-f004:**
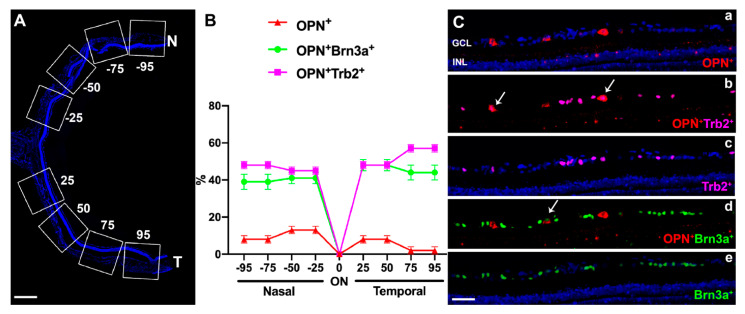
Coexpression of Brn3a, OPN and Tbr2 in cross retinal sections. (**A**) Fluorescent micrograph from a 16 µm thick frozen cross-section on the horizontal naso-temporal axes of a control eye indicating the areas analyzed for counting double labelled retinal neurons at equivalent distances from the optic nerve (ON) to the periphery of the retina (25%, 50%, 75%, and 95%, considering 100% the length of each hemi-retina). N: Nasal, T: Temporal. Scale bar = 1 mm. (**B**) Graphs illustrating the proportions of RGCs that were labelled only with OPN (OPN^+^) or double labelled with Brn3a (OPN^+^Br3a^+^) or Tbr2 (OPN^+^Tbr2^+^) in micrographs obtained at different retinal eccentricities (as depicted in A), from three different cross sections in each of the four control eyes. (**C**). Fluorescence micrograph of a retinal cross section, at approximately 50% from the ON, illustrating immunolabeling with OPN^+^ (red, a), Tbr2^+^ (magenta, c) and Brn3a^+^ (green, e) as well as the double-labelled OPN^+^Tbr2^+^ (b) and OPN^+^Brn3a^+^ (d). Arrows point to double-labelled cells. The section was counterstained with DAPI. GCL, ganglion cell layer. INL, inner nuclear layer. Scale bar = 50 µm.

**Figure 5 ijms-22-11815-f005:**
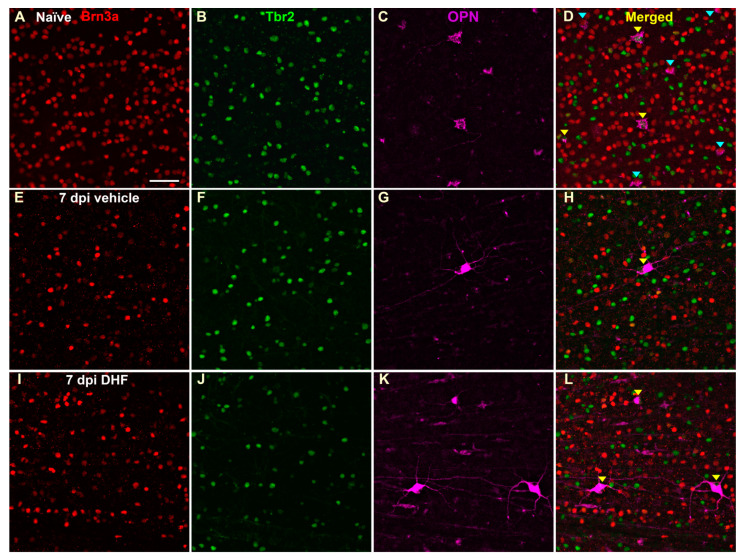
Neuroprotection of axotomized Brn3a^+^, OPN^+^ and OPN^+^Tbr2^+^ RGCs with systemic DHF. Fluorescent micrographs from flat mounted adult albino rat retinas showing brain-specific homeobox/POU domain protein 3A positive retinal ganglion cells Brn3a^+^RGCs (Brn3a) (**A**,**E**,**I**), T-box brain 2 Tbr2^+^ positive RGCs (Tbr2) (**B**,**F**,**J**), osteopontin (OPN) positive OPN^+^RGCs (**C**,**G**,**K**), and merge of these signals (**D**,**H**,**L**) in contralateral control (**A–D**) and in experimental retinas with intraorbitally transected optic nerve treated daily with i.p. vehicle (7 dpi vehicle; **E–H**) or 5 mg/kg DHF (7 dpi DHF; **I–L**) from one day before IONT until processing, analyzed seven days later. Brn3a and Tbr2 labels cell nuclei, while OPN allows to see cell somata as well as primary dendrites on the plane of focus. In the overlapped images of the control retinas, (**D**) one can appreciate the larger densities of Brn3a^+^RGCs compared to OPN^+^RGCs and to OPN^+^Tbr2^+^RGCs. Yellow arrowheads indicate double labelled OPN^+^Tbr2^+^, while blue arrowheads indicate OPN^+^Brn3a^+^. Note that seven days after IONT there are greater numbers of Brn3a^+^RGCs, OPN^+^RGCs and OPN^+^Tbr2^+^RGCs in the DHF-treated retinas when compared to the vehicle-treated retinas, whereas OPN^+^Brn3a^+^ were almost absent irrespective of treatment. Scale bar = 50 µm.

**Figure 6 ijms-22-11815-f006:**
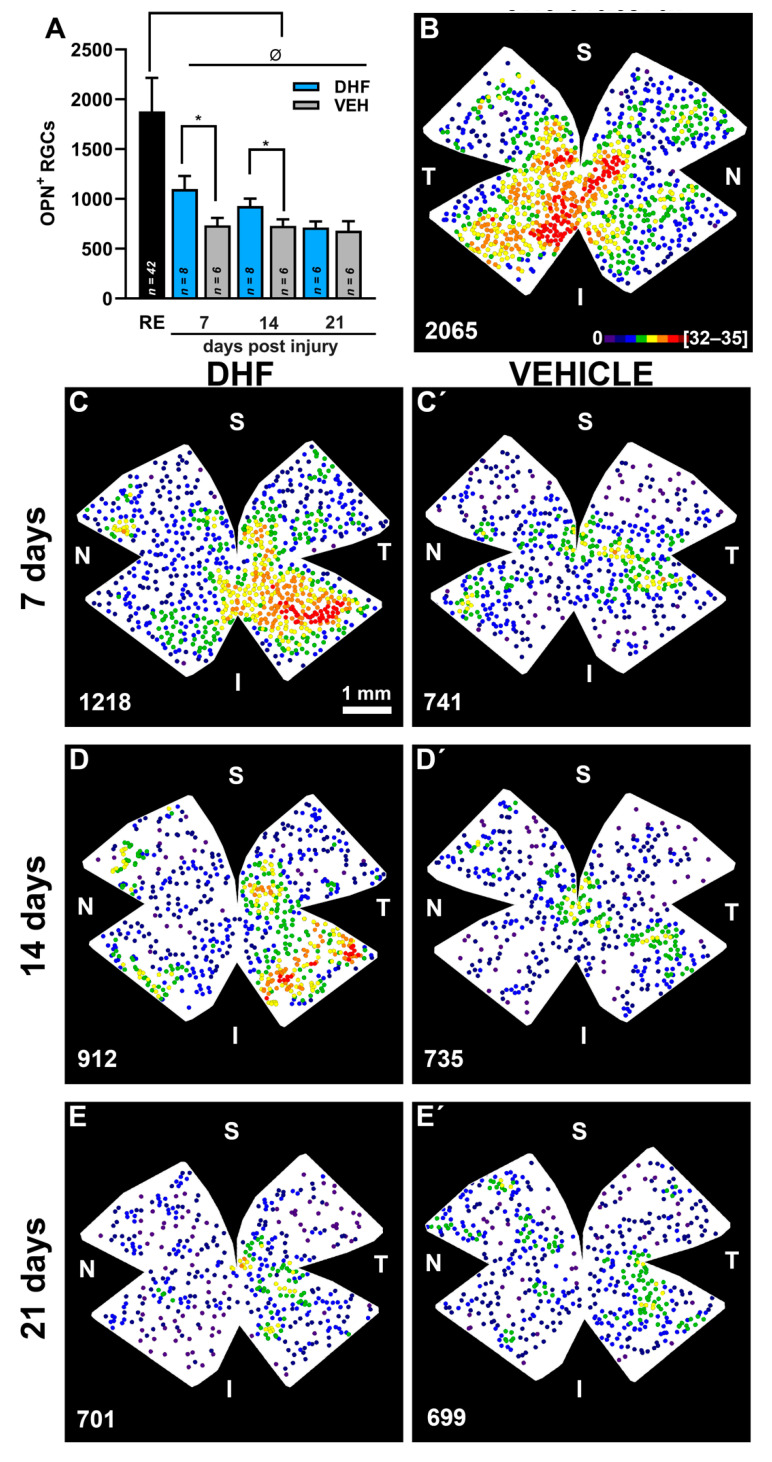
DHF prevents IONT-induced OPN^+^RGC loss. In adult albino rats, the left optic nerve was intraorbitally transected and animals were treated i.p. daily with vehicle (VEH) (1% DMSO in 0.9% NaCl) or DHF (5 mg/kg) starting from one day before IONT until processing, and analyzed at 7, 14 or 21 days to determine survival of RGCs labelled with Osteopontin (Opn^+^RGC). (**A**) Bar histogram showing mean total numbers of OPN^+^RGCs in right fellow (RE) and the loss and neuroprotection of OPN^+^RGCs in left experimental retinas of rats treated with DHF or VEH (*n* for each experimental group analyzed at different days post injury are indicated in the corresponding bar of the histogram). One-way ANOVA * *p* < 0.05, differences between DHF and VEHICLE treated group. One-way ANOVA ^Ø^ *p* < 0.05, differences between control and the experimental groups. (**B–E’**) Representative neighbor maps showing Opn^+^RGC topography in an untouched contralateral retina (**B**), and in retinas treated with DHF (**C**,**D**,**E**) or VEHICLE (**C’**,**D’**,**E’**), analyzed at 7 (**C,C’**), 14 (**D**,**D’**) or 21 (**E**,**E’**) days. Isodensity maps illustrate the neuroprotective effects of DHF over the population of Opn^+^RGCs when compared to vehicle at seven and 14 days. Below each map is indicated the total number of OPN^+^RGCs counted for that retina. S: superior, N: nasal, I inferior, T, temporal. Neighbor maps color scale from 0 to 4 (purple) to ≥32–35 (red) neighbors in a radius of 0.276 mm. Scale bar 1 mm.

**Figure 7 ijms-22-11815-f007:**
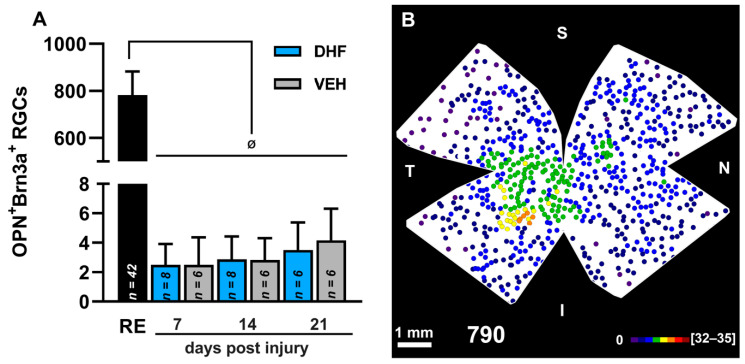
IONT induces massive loss of OPN+Brn3a+RGCs (α-OFF like RGCs). In adult albino rats, the left optic nerve was intraorbitally transected and animals were treated i.p. daily with vehicle (VEH) (1% DMSO in 0.9% NaCl) or DHF (5 mg/kg) starting from one day before IONT until processing, and analyzed at 7, 14 or 21 days to determine survival of RGCs double labelled with Osteopontin and Brn3a (OPN^+^Brn3a^+^RGCs). (**A**) Bar histogram showing mean total numbers of OPN^+^Brn3a^+^RGCs in right fellow (RE) and their massive loss in left experimental retinas of rats treated with DHF or VEH (*n* for each experimental group analyzed at different days post injury are indicated in the corresponding bar of the histogram). One-way ANOVA ^Ø^
*p* < 0.05, differences between control and the experimental groups. (**B**) Representative neighbor map showing OPN^+^Brn3a^+^RGCs topography in an untouched contralateral retina. Below the map is indicated the total number of OPN^+^Brn3a^+^RGCs counted for that retina. S: superior, N: nasal, I inferior, T, temporal. Neighbor maps color scale from 0 to 4 (purple) to ≥ 32–35 (red) neighbors in a radius of 0.276 mm. Scale bar 1 mm.

**Figure 8 ijms-22-11815-f008:**
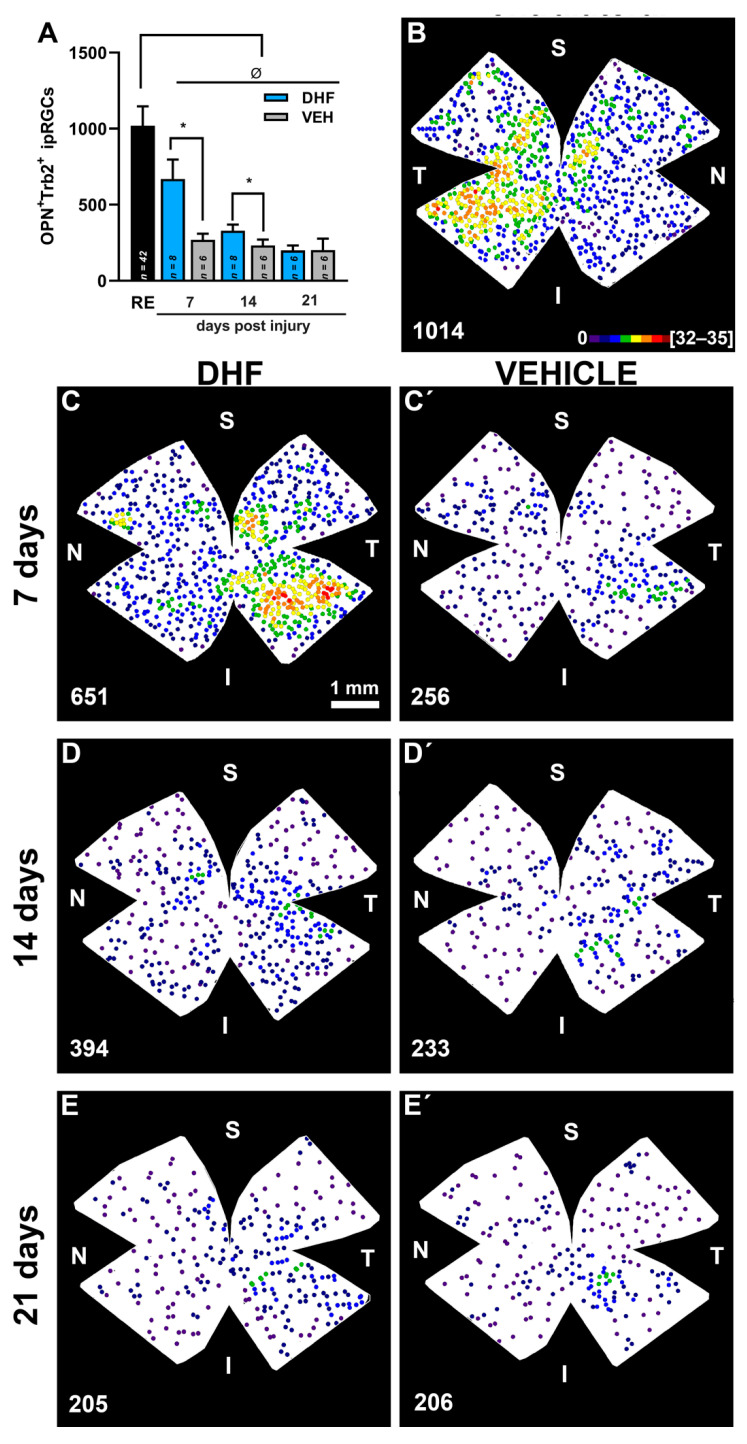
DHF prevents IONT-induced M4-like ipRGCs loss. In adult albino rats, the left optic nerve was intraorbitally transected and animals were treated i.p. daily with vehicle (VEH) (1% DMSO in 0.9% NaCl) or DHF (5 mg/kg) from one day before IONT until processing and analyzed at 7, 14 or 21 days to determine survival of RGCs labelled with Tbr2 and Osteopontin (Tbr2^+^Opn^+^RGC) to identify M4-like/alpha ON sustained RGCs. (**A**) Bar histogram showing mean total numbers of Opn^+^Tbr2^+^RGCs in right fellow (RE) and the loss and neuroprotection of Tbr2^+^Opn^+^RGCs in left experimental retinas of rats treated with DHF or VEH (n for each experimental group analyzed at different days post injury are indicated in the corresponding bar of the histogram). One-way ANOVA ^Ø^
*p* < 0.05, differences between control and experimental groups. One-way ANOVA * *p* < 0.05, differences between DHF and VEH treated group. (**B–E’**). Representative neighbor maps showing Trb2^+^OPN^+^RGCs topography in an untouched contralateral retina (**B**), and in retinas treated with DHF (**C–E**) or VEHICLE (**C’,D’,E’**), analyzed at seven (**C,C’**), 14 (**D,D’**) or 21 (**E,E’**) days. Isodensity maps illustrate the neuroprotective effects of DHF over the population of OPN^+^RGCs when compared to vehicle at seven and 14 days. Below each map is indicated the total number of Tbr2^+^OPN^+^RGCs counted for that retina. S: superior, N: nasal, I inferior, T, temporal. Neighbor maps color scale from 0 to 4 (purple) to ≥32–35 (red) neighbors in a radius of 0.276 mm. Scale bar 1 mm.

**Figure 9 ijms-22-11815-f009:**
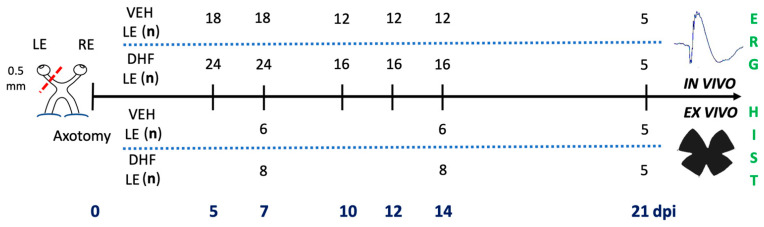
Experimental design: animal groups, timeline, and analyses. Rats received an intraorbital optic nerve section of the left eye on day 0, and were treated daily with intraperitoneal injections of DHF (5 mg/kg) or vehicle (1% DMSO in 0.9% NaCl) starting the day before IONT. In vivo, a longitudinal functional study was performed with full-field electroretinograms at 5, 7, 10, 12, 14 and 21 days post injury (dpi). Ex vivo, retinas were prepared as wholemounts and analyzed with immunohistochemical techniques to assess retinal ganglion cell survival at 7, 14 and 21 dpi.

**Table 1 ijms-22-11815-t001:** Numbers of Brn3a^+^RGCs, OPN^+^RGCs, OPN^+^Trb2^+^RGCs and OPN^+^Brn3a^+^RGCs (mean ± SD; Standard Deviation) in contralateral (RE) and experimental retinas analyzed 7, 14 and 21 days after intraorbital transection of the left optic nerve and daily i.p. treatment with vehicle or DHF (5 mg/kg) from one day before optic nerve injury until processing. One-way ANOVA * Significant when compared with contralateral retinas in the same RGC type (*p* < 0.001). One-way ANOVA ^†^ Significant when compared with vehicle-treated retinas in the same RGC type at the same time point (*p* < 0.05). One-way ANOVA ^‡^ Significant when compared with the previous time point in the same RGC type (*p* < 0.05). RE, contralateral right retinas. 7 d, 14 d, 21 d, experimental left retinas.

	Brn3a				OPN				OPN-Trb2				OPN-Brn3a			
	RE	7 d	14 d	21 d	RE	7 d	14 d	21 d	RE	7 d	14 d	21 d	RE	7 d	14 d	21 d
**RE (*n*)**	42				42				42				42			
Mean	81,085				1917				1015				782			
SD	3056				229				120				100			
**VEH (n)**		6	6	5		6	6	5		6	6	5		6	6	5
Mean		37,121 *	22,754 *^‡^	8626 *^‡^		735 *	729 *	679 *		269 *	233 *	202 *		3 *	3 *	4 *
SD		4312	4739	1295		73	64	97		43	39	75		2	1	2
%	100	45.7	28.1	10.6	100	38.4	38.0	35.4	100	26.5	22.9	19.9	100	0.4	0.4	0.5
**DHF (*n*)**		6	6	5		6	6	5		6	5	5		6	5	5
Mean		76,051	52,620 *^†‡^	12,089 *^†‡^		1115 *^†^	956 *^†‡^	714 *^‡^		669 *^†^	341 *^†‡^	205 *^‡^		3 *	3 *	4 *
SD		4312	4581	1591		148	60	60		127	41	32		1	2	2
%	100	93.8	64.9	14.9	100	58.1	49.8	37.2	100	65.9	33.7	20.2	100	0.4	0.4	0.5

## Data Availability

The data presented in this study are available on request from the corresponding author.

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
