# Peer review of "7,8-Dihydroxiflavone Maintains Retinal Functionality and Protects Various Types of RGCs in Adult Rats with Optic Nerve Transection"

_ijms, 2021, doi:10.3390/ijms222111815_

Round 1

Reviewer 1 Report

Gallego-Ortega, A. and colleagues presented an interesting group of results on the neuroprotective effects of 7,8-Dihydroxiflavone. The authors used the intraorbital optic nerve section as an experimental model of ganglion cell degeneration. Nerve section led to loss of neuronal activity. Systemic treatment of 7,8-Dihydroxiflavone through intraperitoneal injection improved neuronal function. Improvement in neuronal activity was associated with reduced ganglion cell loss, suggesting a direct effect of 7,8-Dihydroxiflavone in protecting retinal ganglion cells against neurodegeneration.

The results are clear. Attention must be giving to the fact that the research group recently published a similar set of results on the neuroprotective effects of 7,8-Dihydroxiflavone on Brn3a positive ganglion cells degeneration using the same experiment system (PMID: 34245756). In addition, I have major concerns in relation to the mechanisms of action of the 7,8-Dihydroxiflavone and the manuscript organization.

Manuscript organization

The organization of the manuscript is confusing and must be improved. The main message of the manuscript is highlighted in the title and is based on the effects of 7,8-Dihydroxiflavone on neuroprotection. However, the authors spend almost half of the manuscript describing the characterization of different ganglion cell populations in the rat retina, which they separate into three distinct populations based on immunofluorescence staining Brn3a+, OPN+ and OPN+Tbr2+. The information is relevant, however, in the present form, it makes the manuscript confusing and hard to read. The manuscript needs to be organized in a more comprehensive way. My suggestion is to reduce the attention giving to the description of the ganglion cell population and focus on its relevance to the action of 7,8-Dihydroxiflavone.

How 7,8-Dihydroxiflavone protects against neurodegeneration

In the manuscript, the authors perform systemic treatment of 7,8-Dihydroxiflavone. The improvement in neuronal function can be caused by a plethora of systemic effects that do not relate to the neuroprotective action of 7,8-Dihydroxiflavone as a TrkB agonist in ganglion cells. How does this drug produce the neuroprotective effect? Because the authors just published similar results in another manuscript (PMID: 34245756) I think that providing mechanistic data on how 7,8-Dihydroxiflavone protect against ganglion cell degeneration is essential to move the field forward.

Translational relevance

As highlighted above, the effects of 7,8-Dihydroxiflavone on improving neuronal function and degeneration are clear. However, the results are poorly discussed. The authors should discuss the relevance of their finds to human health. Human diseases that cause ganglion cell neurodegeneration (e.g. glaucoma) should be highlighted in this context.

Author Response

Dear Reviewer, 

You can see the comments to your requests in the attached file

Thank you in advance,

Regards

Reviewer 2 Report

In this article, Gallego-Ortega et.al., shows that IONT resulted in massive progressive loss of Brn3a+RGCs and in massive but not progressive loss of OPN+RGCs and OPN+Tbr2+RGCs suggesting a certain resilience of these RGC types to axotomy. Moreover, DHF-treatment in adult rats with IONT resulted in significant protection for the Brn3a, OPN and OPNTbr2 RGCs, suggesting that all these populations are responsive to DHF-induced neuroprotection. The manuscript is well written, and results are well presented. This article is a valuable addition to the field and has interesting observations which are beneficial to researchers in the areas of retinal neurodegeneration and other retinal associated complications. The reviewer does not notice significant limitations of the manuscript apart from the concerns discussed below.

Results
Table 1 – Fix the alignment of the table as name of the markers seems to be misaligned. Also it will be good to have colour patterns in the table for each marker.
Line 181 – Delete the repeated “10”
Line 286 – Replace “tupe” with “type”.
Line 294 – Replace “reinal” with “retinal”
-For all the figure captions, add appropriate n numbers and also mention the analysis used.

Discussion
-Discuss further on the 5mg/kg DHF concentration in clinical use and also discuss other ways of administration if possible. 

Author Response

(The authors gave the same response as above.)

Reviewer 3 Report

The paper entitled: “7,8-Dihydroxiflavone maintains retinal functionality and protects various types of RGCs in adult rats with optic nerve transection” by Alejandro Gallego-Ortegaet al. deals with the study the effect of 7,8-Dihydroxiflavone on retinal functionality in rats with optic nerve transection.

Although the paper addresses an issue of interest in the field, the authors may wish to consider the following prior to publication.

Major concern:

The authors used the just one dose of 7,8-Dihydroxiflavone (5 mg/kg, ip). Why?

A dose-ranging should be necessary, unless the authors have a good reason to use one specific dose.

Even though the paper presents some limitation (e.g. IHC), the data generated are robust.

Introduction Section: the authors wrote: “Several strategies have been explored to slowdown the loss of RGCS following several types of retinal insults, including the use of alpha two agonists  [14–21] and the use of substances with neuroprotective properties, such as trophic factors and neurotrophins [22–29] of which BDNF is the most potent neuroprotectant for the retina [30], and the caspase inhibitors [31], among others [32].” in this Section, the authors should report the relevant and recent papers showing new evidence of RGCs protection by different pharmacological approaches, this will be useful to the readers that have not familiar with the eye field (please report these relevant and recent papers: 1) J. Neuroinflammation. 2021 Sep 16;18(1):206. doi: 10.1186/s12974-021-02263-3.PMID: 34530842; 2) Front Pharmacol. 2021 Jul 21;12:705405. doi: 10.3389/fphar.2021.705405. eCollection 2021.PMID: 34366858; 3) Biochem Pharmacol. 2020 Oct;180:114199. doi: 10.1016/j.bcp.2020.114199. Epub 2020 Aug 13. PMID: 32798466).

Introduction Section: please add at the end of this section a clear sentence/s on the aim of the study.

I suggest adding a brief paragraph with Conclusion to wrap-up the salient points and the meaning of the data generated.

Author Response

(The authors gave the same response as above.)

Round 2

Reviewer 1 Report

The authors addressed the issues in the manuscript and the text should be accepted for publication.

Author Response

The authors thank the reviewer for his kind comments.

Reviewer 2 Report

Gallego-Ortega et.al., have thoroughly addressed all the concerns raised in the initial article and the efforts are greatly appreciated. The revised research article can deserve publication without further changes.

Author Response

(The authors gave the same response as above.)

Reviewer 3 Report

the revised version is ok now and can be published. There are some typos in the text, please check the text (e.g. on line 100).

Author Response

The authors would like to thank the reviewer for his comments. We have looked into the text and corrected all typos.